# Polymerase Epsilon-Associated Ultramutagenesis in Cancer

**DOI:** 10.3390/cancers14061467

**Published:** 2022-03-12

**Authors:** XuanXuan Xing, Ning Jin, Jing Wang

**Affiliations:** 1Department of Cancer Biology and Genetics, James Comprehensive Cancer Center, The Ohio State University Wexner Medical Center, Columbus, OH 43210, USA; xuanxuan.xing@osumc.edu; 2Division of Medical Oncology, Department of Internal Medicine, James Comprehensive Cancer Center, The Ohio State University Wexner Medical Center, Columbus, OH 43210, USA

**Keywords:** polymerase epsilon, ultramutagenesis, cancer, prognosis, immunotherapy

## Abstract

**Simple Summary:**

DNA polymerase epsilon is implicated to play a major role in DNA synthesis of the leading strand. In some cancer types, especially colorectal and endometrial cancers, polymerase epsilon is mutated at several hotspots, causing large amounts of mutations, termed ultramutation. The aim of this article is to describe the characteristics of polymerase epsilon mutations including their mutation sites and signatures, elucidate the underlying mechanisms of its ultramutagenesis, discuss its good prognosis and favorable responses to immunotherapies, and speculate on possible strategies to improve treatment of ultramutated cancers.

**Abstract:**

With advances in next generation sequencing (NGS) technologies, efforts have been made to develop personalized medicine, targeting the specific genetic makeup of an individual. Somatic or germline DNA Polymerase epsilon (*PolE*) mutations cause ultramutated (>100 mutations/Mb) cancer. In contrast to mismatch repair-deficient hypermutated (>10 mutations/Mb) cancer, PolE-associated cancer is primarily microsatellite stable (MSS) In this article, we provide a comprehensive review of this *PolE*-associated ultramutated tumor. We describe its molecular characteristics, including the mutation sites and mutation signature of this type of tumor and the mechanism of its ultramutagenesis. We discuss its good clinical prognosis and elucidate the mechanism for enhanced immunogenicity with a high tumor mutation burden, increased neoantigen load, and enriched tumor-infiltrating lymphocytes. We also provide the rationale for immune checkpoint inhibitors in PolE-mutated tumors.

## 1. Introduction

Colorectal cancer (CRC) is a very common cancer worldwide. In the United States, CRC is the third most diagnosed cancer in men and women, and the second-leading cause of cancer deaths when men and women are combined. It is estimated that there were 149,500 new cases of CRC and 52,980 deaths in 2021. While the localized CRC tumor is largely curable with surgery followed by radiotherapy, in some cases metastatic CRC is treated by chemotherapy and immunotherapy. More and more patients benefit from precision medicine, based on a molecular genetic diagnosis.

Previously explored precision treatments have focused on targeting oncogenes (e.g, *KRAS*, *PIK3CA*, and *BRAF*). Currently, the attention toward CRC has shifted to clonal–stromal–immune perspectives, highlighting tumor immunity microenvironment heterogeneity.

Transcriptomic analysis identified four consensual molecular subtypes (CMS) of CRC: CMS1—microsatellite instability (MSI-H)/immunogenic tumors, accounting for 14% of all CRC cases and hypermutated with strong immune activation; CMS2—canonical tumors (37% of CRC cases), epithelial and exhibiting MYC and WNT signaling pathway activation; CMS3—metabolic (13% of CRC cases), epithelial, and displaying evident metabolic dysregulation; and CMS4—mesenchymal (23% of all CRC) featuring stromal invasion, transforming growth factor-beta activation, and angiogenesis [1]. DNA mismatch repair (MMR) defects lead to the CMS1 subset of CRC and the hypermutated phenotype. Immune checkpoint inhibition is a new therapeutic strategy that has shown promising efficacy in many cancer types. Significant immunity response and prolonged progression-free survival (PFS) were observed in MMR-deficient metastatic CRC [2].

Notably, another type of so-called “ultramutated” CRC, with more than 100 mutations/Mb, also presents a high neoantigen load, tumor-infiltrating lymphocytes, and predicts a favorable response to immune checkpoint inhibition [3]. Different from MMR-deficient hypermutated CRC, several unique point mutations in the proofreading domain of replicative DNA polymerase epsilon (*PolE*) were found in these CRC and the tumor is characterized as being microsatellite stable (MSS) [4].

In this review, we provide a comprehensive overview of this PolE-associated ultramutated tumor. We describe its molecular characteristics, including the mutant sites in *PolE*, the mutation signature of this type of tumor, and the progress in the molecular mechanism of its ultramutagenesis and carcinogenicity. We describe the clinical features and explain the potential reasons for its good prognosis, i.e., tumor-infiltrating lymphocytes and error threshold. Finally, we provide histological and clinical data on its immunity checkpoint blockade, addressing the favorable benefit from anti-PD-1 immunotherapy. 

## 2. *PolE* Mutations Generate Ultramutation Genotype in Cancer

Genome instability is one of the hallmarks of cancer. In CRC, DNA MMR defects caused by mutations in repair enzymes have been found to be responsible for a high level of MSI-H and germline mutations of MMR genes are responsible for Lynch syndrome, also called hereditary nonpolyposis colorectal cancer (HNPCC) [5,6]. Clinically, CRC patients with MSI-H tumors have better prognoses and show more response to immunotherapy than those with MSS tumors [2,7]. On the other hand, although Larry Leob and his colleagues hypothesized in the 1970s that defective DNA polymerases could lead to genome instability [8], there were few reports of mutations in DNA polymerases [9,10,11,12].

In 2012, data from a comprehensive genomic study of colorectal carcinoma were made available by The Cancer Genome Atlas (TCGA), including exome sequencing of 224 tumor samples. The data show that while most hypermutated tumors (>10 mutations/Mb) exhibited MSI-H or MMR deficiency, a separate class of ultramutated tumors (>100 mutations/Mb) is surprisingly MSS and bears somatic mutations in *PolE* [4], which is responsible for DNA synthesis in the leading strand during DNA replication [13]. However, somatic mutations in the *PolD1* gene encoding the catalytic subunit of Pol δ, another DNA replicative polymerase responsible for DNA synthesis in the lagging strand [14], are not detected in ultramutated tumors. Instead, they are found in sporadic hypermutated tumors together with MMR defects [15]. Pol δ not only proofreads the errors generated by itself in the lagging strand, but also extrinsically corrects the errors produced by Polε in the leading strand [16]. Soon after, germline mutations in *PolE* were reported to create a predisposition to colorectal adenomas and carcinomas [17]. The mutations are mainly located in the proofreading exonuclease domain of *PolE* [15]. P286R and V411L are the two most common somatic mutations in *PolE*. Additionally, P286H/S, F367S, S459F, and other mutations also appear in sporadic tumors but at a lower frequency, whereas the V424L mutation occurs in germline tumors [15]. While most of these mutations affect highly conserved residues in or adjacent to the exonuclease motifs at the DNA-binding interface, V411L is located beyond the exonuclease catalytic core [15,18]. Notably, the vast majority of *PolE* mutations are heterozygous [15]. A genetic study in yeast showed that, except for V411L, a vast majority of *PolE* mutations at the DNA binding cleft have elevated spontaneous mutation rates with varying degrees [19]. *PolE* mutations are most frequently found in colorectal cancer (1–3%) and endometrial cancer (6–12%), and rarely found in lung, breast, brain, prostate, kidney, ovarian, bone, and gastric cancers [3]. In addition, other somatic point mutations in the polymerase domain of *PolE* (R567C, K593C, S595P, E611K, and L621F) and a deletion frameshift in the middle region of the PolE gene (V1446fs *3) are possible pathogenic factors in cancer [20,21]. Mutations in non-catalytic subunits of Polε, *PolE2*, *PolE3* and *PolE4*, are rare in cancer and their involvement in ultramutagenesis is not clear.

Studies in yeast and mouse models mimicking human *P286R* mutation supported the hypothesis that ultramutation observed in tumors was driven by a single base alteration in human *PolE* [22,23]. This analogous substitution in yeast yielded the most powerful known *PolE* mutator phenotype, which twice surpassed that of proofreading-deficient *PolE* mutants [22]. Two very recent studies explored how *PolE-P286R* affects polymerase function [24,25]. Shcherbakova’s lab showed that yeast *Pol2-P301R* mutant polymerase has increased activation on standard DNA templates, as well as an increased ability for bypassing hairpin structures and extension. A mispaired primer termini mutant Arg residue, as demonstrated by the crystal structure, enters the primer binding cleft of the exonuclease active site which then blocks primer partitioning [25]. The authors hypothesized that the enzyme is kept in a state of hyperactive polymerization by the mutant, which results in mutagenesis. However, purified Polε-P301R displays a relatively low error rate in vitro, as compared to exonuclease-deficient Pol2, suggesting that additional factors must enhance the impact of its effect on ultra-mutagenesis in vivo [24]. MMRs and the proofreading of DNA polymerase δ prevent catastrophic errors made by *Pol2-P301R* in vivo [26]. Similarly, monoallelic PolE proofreading deficiency in *MLH-1* (a human MMR gene) deficient *HCT-116* cells caused a rapid increase in mutations, while the re-introduction of wild-type *MLH-1* suppressed the mutation accumulation [27]. Deletion of translesion synthesis (TLS) polymerases, *Pol**κ*, suppressed the ultramutation generated by *PolE-P286R* in a fission yeast model [28]. Shcherbakova’s lab also characterized 13 other cancer-associated PolE mutations in the yeast model and found that only mutations that directly alter the exonuclease domain of the DNA binding cleft increase the mutation rate. Of these, the frequently occurring mutants were more powerful mutators than the less commonly occurring ones, which supports the idea that the mutator phenotype plays a causative part in tumorigenesis [19]. Whole exome sequencing (WES) data in yeast also showed similar results [29,30].

The heterozygous and homozygous proofreading deficient mice, *PolE^exo−/+^* and *PolE^exo−/−^*, were constructed with alanine substitutions at residues D272 and E274. Approximately half of *PolE^exo−/−^*, but not *PolE^exo−/+^,* mice developed adenocarcinomas in the small intestine [31]. In comparison, heterozygous *PolE^P286R/+^* mice are highly tumorigenic, producing tumors in multiple organs including the lung, thymus, shoulder, leg, and ovary [23]. An estimate of 1.6 (average of 0.9, 1.6, and 2.1 in three studied MEF cell lines) nucleotide substitutions per Mb per cell division was observed by whole genomic sequencing (WGS), which is at least 3 orders of magnitude higher than the intrinsic DNA replication error rate. WGS of seven primary tumors from *PolE^P286R/+^* mice, including three lung adenocarcinomas, three T-cell lymphomas, one cutaneous squamous cell carcinomas, and four tumors from two *Pole^P286R^ LSL-PolE^P286R^* (hemizygous) mice, revealed a very high frequency of mutations, in the range of 10 to 100 mutations/Mb [23]. *PolE^P286R/+^* is reported to be the first genetically engineered mouse model to repeat the clonal variation, high mutation burden, and heterogeneity characteristics of human cancer [32].

## 3. Mutational Signatures of *PolE*-Associated Tumors

In 2013, Alexandrov et al. discovered over twenty distinct mutational signatures upon analyzing 4,938,362 mutations from 7042 tumors. The tenth signature among these was linked to PolE mutations found in colorectal cancer and endometrial cancer [33]. This type of signature, termed single base substitution 10 (SBS10), is not transcriptional strand biased and the landscape of this mutational signature is characterized predominately by TCG→TTG (>20%) transitions, TCT→TAT (>20%) transversions, and TTT→TGT (~7%) transversions, three subtypes named SBS10a, SBS10b and SBS28, respectively [33,34]. The frequency of TCT→TAT mutation is slightly higher in *PolE V411L* than in *PolE P286R* [35]. Japanese researchers explored the WES of 2141 solid tumor samples derived from 2042 cancer patients [36]. Their work confirmed the landscape of the mutational signature of *PolE* mutations with regard to number and frequency. These are, in descending order, TCT→TAT, TCG→TTG, and TTT→TGT. Further, they analyzed the mutation patterns of individual somatic tumors with different *PolE* mutations, including P286R (six samples), P286R&L120I (one sample), V411L (one sample), A189D&F367V (one sample), and K717N (one sample). Tumors with P286R or V411L mutations showed a mutation pattern consistent with that of PolE-associated tumors, although their genetic characteristics were found to be dramatically different in yeasts [19]. Interestingly, the pattern described above was enriched in *SACS, LRP2, WDR87,* and *XIRP2* genes, but not in *POLE* and *PTEN*. The *HCC2998* colon cancer cell line, harboring a *PolE-P286R* mutation, also showed a disproportionally large increase in GC→TA transversion, with a particular preference for AGA/TCT sequence context [15]. In vitro, a *lacZ* forward mutation assay with purified N terminal 1–1189 of the catalytic subunit of mutant human *PolE-P286R* showed a bias for TCT→TAT transversion over the complementary AGA→ACA transversion [37]. Different from somatic *PolE* mutations, ultramutated glioblastoma with the germline *PolE V424L* mutation is predominated by C > G transversion [38].

In yeast models, the mutation spectrum of the *CAN1* forward assay and the *lacZ* forward assay showed an increase in the GC→AT transition/GC→TA transversion ratio, supporting the idea that mutations in *Pol2-P301R* mutant cells were produced by the Polε-P301R protein during DNA replication. Notably, the predominant base substitution in yeast is GC→AT transition, whereas it is GC→TA transversion in human tumors, possibly due to the slight difference in the exonuclease domain of *PolE* between yeast and humans [24]. 

## 4. Altered Pathways and Responses to Chemotherapy in PolE-Associated Cancer 

Since point mutations in PolE produce ultramutation genotypes in tumors, one question is whether these mutations occur in specific oncogenes and/or tumor suppressor genes. Nebot-Bral and his colleagues analyzed altered oncogenic pathways in ultramutated EC and CRC, finding that the TP53 and TGF-β pathways are inactivated whereas the WNT, PI3K and RTK/RAS pathways are activated [3], not significantly different from MMR deficient CRC/EC tumors. However, unique alterations in certain pathways were found in PolE-associated tumors [36]. For example, dysfunction of the PTEN pathways and the activation of double-strand break repair, homologous recombination (HR), and non-homologous end-joining (NHEJ) pathways specifically occurred in *PolE*-associated ultramutated tumors, but not in hypermutated MSI-H tumors. A recent study also reported truncations of the homologous recombination repair gene, *BRCA1/2*, in *PolE* mutated CRC and EC [39,40]. Another study found that truncated *TP53 R213** was specifically enriched in *PolE P286R* mutant CRC [35]. Additionally, there are two other pathways specifically mutated in *PolE*-associated tumors: the transcriptional regulation pathway, mediated by hypoxia inducible factor 1 alpha (HIF1α), and the p160 steroid receptor co-activator (SRC) pathway [36]. The mechanisms of the dysregulation of these pathways are not clear and need further investigation.

Chemotherapy is the primary treatment for CRC patients. Van Gool and colleagues investigated whether mutated *PolE* changes the sensitivity to chemotherapeutic agents in mouse-derived embryonic stem (mES) cells. Two nucleoside analogs, cytarabine and fludarabine, were found to show a higher potency in inhibiting cell viability in cell lines with the *PolE* mutations P286R, S297F, and V411L than in the wild-type *PolE* cell line. However, other chemotherapeutic agents, 5-fluorouracil, cisplatin, paclitaxel, doxorubicin, etoposide, and methotrexate, did not display any difference in efficacy between the wild type and mutant PolE cell lines [41]. Mechanistically, exonuclease deficient Polε efficiently incorporates cytarabine but has difficulty extending it, resulting in a high sensitivity to cytarabine-induced cellular cytotoxicity. Additionally, the Vitamin B6 compound, pyridoxal (PL), converts to pyridoxal 5′-phosphate, inhibits PolE-mediated DNA synthesis by competing with nucleotide substrates and thus decreases the cell viability of HeLa cells [42]. These results indicate that nucleoside analogs, which inhibit DNA polymerases, have the potential to treat *PolE*-associated tumors effectively. Sulphoquinovosyl diacylglycerol (SQDG) was reported as the first mammalian PolE-specific inhibitor [43], which might be a potential drug candidate.

## 5. Good Prognosis in Patients Harboring PolE Mutations

It has been shown that *PolE* mutations are identified in pre-malignant lesions in EC and CRC with prominent CD8^+^ T-cell infiltration. It suggests that somatic *PolE* mutations are early events and may contribute to tumor initiation and good prognosis. *PolE*-associated ultramutated CRC and EC and high-grade gliomas have been shown to have good prognoses [42,43,44,45], although mutation rates in those cancers are mostly higher than 100 per Mb.

In univariable analysis, *PolE*-mutated EC patients have significantly improved outcomes in terms of PFS, disease-specific survival (DSS), and overall survival (OS), as compared with patients with wild-type *PolE*. However, in multivariable analysis, *PolE* mutations are only significantly associated with improved PFS. Although no definite conclusions as to the effects of adjuvant treatment on *PolE*-mutated cases can be drawn, a meta-analysis showed that *PolE* mutations were associated with disease-free survival (DFS) and improved progression-free survival (PFS), featuring pooled hazard ratios of 0.34 (95% confidence interval (CI), 0.15–0.73) and 0.35 (95% CI, 0.13–0.92) in treated and untreated patients, respectively [46].

Another study on ECs also showed that patients with PolE mutations have improved PFS (100% after 90 months), as compared to PFS of patients with other types of tumors (<80% after 50 months) [42]. In addition, a European research group reported that women with PolE-associated ECs have fewer recurrences (6.2% vs. 14.1%) and EC deaths (2.3% vs. 9.7%) than other ECs [43]. Other studies have also shown similar findings [47,48,49,50,51,52].

Colorectal cancer patients with PolE mutations are typically diagnosed younger than patients with wild-type PolE (median of 54.5 years vs. 67.2 years, *p* < 0.0001) and at an earlier disease stage (*p* = 0.006, x^2^ test for trend). In addition, *PolE* mutations are more common in male than female patients and occur more frequently in the right colon than the left [44]. In stage II/III tumors, only four of 59 (8%) PolE-mutated patients experienced recurrence after surgery (median duration of 4.7 years), compared to 130 of 689 patients (18.9%) with MSI-H CRC and 1074 of 3905 patients (27.5%) with MSS CRC. In stage II patients, one recurrence out of 34 cases with PolE mutations (2.9%) was noted, compared to 61 of 426 MSI-H (14.3%) and 360 of 1735 MSS (20.7%) cases. This indicates that PolE mutations are associated with a greatly reduced risk of recurrence. For stage III diseases, however, the effect of PolE mutations on recurrence was insignificant [44]. Other studies also showed that CRC patients with PolE mutations have favorable prognoses [53,54,55,56,57,58,59]. Interestingly, one study reported that all the patients who were diagnosed with *PolE*-mutated CRC at younger than 50 years of age were found to harbor P286R mutation [58].

Together, these studies indicate that cancer patients with PolE mutations have good prognoses and favorable outcomes, in general. However, the underlying mechanisms are not completely understood. One potential mechanism could involve increased immune cell infiltration and enhanced immune surveillance, as shown in Figure 1.

Church et al. investigated the interplay between *PolE*-associated endometrial cancer and immunogenicity [60]. The preliminary data revealed that *PolE*-associated EC often presents a prominent lymphocytic infiltration and lymphocytic reaction similar to Crohn’s disease. Later studies showed that CD8^+^ T cells were more enriched in the center and at the invasive tumor front, in comparison with the MSI-H or MSS counterparts in endometrial cancer. T-cell infiltration in *PolE*-associated tumors was also enhanced by the expression of CD3 and cytolytic marker TIA-1. TIA-1 expression in CD8^+^ lymphocytes shows that these T cells may contribute to the mediation of anti-tumor immune response. The observation that nearly all tumor infiltrated lymphocytes (TILs) are CD8^+^ in PolE-associated EC tumors increasingly supports TILs’ enhanced antitumor effects [61]. 

Analysis of TCGA data in EC found that PolE-associated tumors display the enrichment of a 200 gene signature which corresponds to T-cell infiltration, including the upregulation of genes in T-cell-mediated cytotoxicity; cytotoxic T-cell differentiation and activation markers; CD8A and IFNγ; Eomes; T-bet; granzymes B, H, K, and M; perforin; and IFNγ-induced cytokines, CXCL9 and CXCL10. Notably, *PolE*-associated endometrial tumors also show upregulation of the T-follicular helper genes, CXCL13 and CXCR5, which were recently shown to be powerfully predictive of a positive outcome in CRC [62]. As a comparison, in MSI-H tumors, only CD8A and IFNγ were upregulated, and no difference of the expression of cytotoxic T-cell differentiation and activation markers was detected [60]. In addition, in some cases, despite limited numbers, expression of cytotoxic T cell markers and IFNγ-induced cytokines (e.g., PRF, GZMH, CXCL 9/10) is significantly higher in tumors bearing *PolE* mutations, when compared to MSI-H tumors [62]. These studies support the conclusion that *PolE*-mutant tumors exhibit higher T-lymphocyte infiltration as compared to other endometrial tumors, and that these lymphocytes have the capacity to show antitumor activity.

In another study, *PolE*-mutant endometrial tumors were shown to have the highest expression of immune-associated genes in comparison to MSI and MSS tumors [63]. Similarly, *PolE*-mutant tumors also exhibit a high expression of T cell markers, including PD-1, CD8A, and cytotoxic T-lymphocyte-associated protein 4 (*CTLA-4*) [63], which may be indicative of T-cell infiltration. In addition, leukocyte subsets, as determined by CIBERSORT (https://cibersort.stanford.edu/, accessed on 31 January 2022), show that *PolE*-mutant endometrial tumors display much higher proportions of CD8^+^ T cells, T helper cells, M1 macrophages, and activated NK cells as compared to MSI-H and MSS tumors.

PolE-mutant EC shows a heightened cytotoxic T-cell response, reflected by more CD8^+^ TILs, alongside an increased expression of T-cell exhaustion markers, which is consistent with chronic antigen exposure [44,60,64]. In CRC, it has been shown that the number of CD8+ cytotoxic TIL in PolE-mutated CRC significantly exceeds that of MMR proficient tumors but is not different from that of MMR deficient tumors. Furthermore, expression of cytotoxic T-cell markers (CD8A, IFNG, and PRF), effector cytokines (CXCL9 and CXCL10), and immune checkpoint proteins (PD-1, PD-L1, and CTLA-4) is significantly increased in *PolE*-mutant CRC [44].

To determine the mechanisms of increased TILs in PolE-associated tumors, many studies evaluated the correlation of antigenic neoepitopes as a consequence of ultramutation in *PolE*-mutant tumors [44,45,46,47]. One study showed that approximately 5.9% (7880/134,473) of missense mutations in ECs were possibly antigenic. Of these, 73% (5767/7880) presented in *PolE*-associated tumors, which indicated a significantly higher number of antigenic mutations when compared with MSI-H or MSS tumors (median of 365.5 vs. 16 and 2 for MSI-H and MSS, respectively). This could contribute to an active T-cell response in those tumors [60]. Similarly, *PolE*-mutant CRC shows an increased neoantigen load and a high degree of TILs, as compared with *PolE*-WT CRC [46]. In addition, the neoantigen load in CRC correlates significantly with CD45RO^+^ memory T-cell density but not with that of CD8^+^ (cytotoxic), CD3^+^ (total), or FOXP3^+^ (regulatory) T cells [46].

Despite being able to mount an anti-tumor immune response, some patients with *PolE* mutation present with bulky tumors. Since there was no significant difference in HLA class I protein expression between *PolE*-mutant and MSI-H tumors, the likelihood of immune escape due to lack of antigen presentation may be low. Instead, there may be suppression of the adaptive immune system that could explain immune evasion in *PolE*-driven EC. Although expression of T-cell exhaustion markers, including LAG3, TIM-3, and TIGIT, and immune checkpoint proteins, PD-1 and CTLA-4, is strongly associated with CD8A expression in all subtypes of ECs, their expression in PolE mutant tumors is significantly and substantially upregulated, indicating prolonged antigen stimulation and increased adaptive immune resistance [60]. Consistent with this study, Church and colleagues demonstrated that PolE-mutant EC tumors express higher levels of the immune checkpoint-related proteins, PD-L1 and PD-L2, than MSI-H and MSS tumors [60].

Taken together, these studies demonstrate that PolE-mutant tumors are associated with increased cytolytic T-cell response and upregulated expression of immunosuppressive checkpoints [47], which may contribute to a good prognosis and an overall favorable outcome for PolE-mutant patients. Although it has been shown in *PolE^P286R/+^* mice that CD3/CD19 thymus-based T cell lymphomas infiltrated the lungs aggressively [23], more studies are needed to establish a causative relationship between PolE mutations, enhanced immune surveillance and good prognosis. In addition, increased expression of immunosuppressive checkpoints in *PolE*-mutant tumors demonstrates an excellent potential to respond to anti-PD1/PD-L1 immunotherapy.

## 6. Response to Immunotherapy in Patients with PolE Mutations

The observations that *PolE*-mutant tumors display a high cytotoxic T-cell response and an upregulated expression of immune checkpoints implicates that patients with *PolE* mutations may likely have a good response to immunotherapy with checkpoint inhibitors. Table 1 summarizes the immunotherapy trials in PolE mutant cancers. 

The first reported immunotherapy in *PolE*-mutant cancer was carried out in non–small cell lung cancer (NSCLC) [48]. In this trial, two patients responded well to an anti-PD-1 antibody pembrolizumab. When examining tumor samples, one sample showed a stop-gain mutation in *PolE* as well as mutations in DNA-dependent protein kinase catalytic subunit (PRKDC or DNA-PK). In the second sample, a deleterious mutation in *PolE4* encoding the third non-catalytic subunit of DNA PolE was detected. Although these mutations were not point mutations in the exonuclease domain of *PolE*, the tumors still showed a high mutation burden and displayed a significant response to the anti-PD-1 antibody.

In a case report, a 53-year-old female EC patient with *PolE-V411L* and nonsense *PolE-R114** mutations showed an exceptional response to pembrolizumab [49]. An estimated total tumor mutation burden (TMB) of 4500 and 6500 nonsynonymous mutations were detected by WES in primary and metastatic tumor samples, respectively. This is consistent with TCGA data which indicate that metastatic/recurrent ECs harboring mutant *PolE* also display a high mutational burden [52].

In another case report, a glioblastoma patient with germline *PolE V424L* mutation showed a good response to immunotherapy with pembrolizumab [38]. All tumors, including one primary and two metastases were ultramutated, with over half of 17,276 to 20,045 mutations nonsynonymous. Subsequent work identified 2040 to 3254 high-quality neoantigenic mutations per sample, in which one of the metastases carried the highest mutation burden. After three weeks of treatment with pembrolizumab, a decrease in the right frontal lesion was observed, but there was a perilesional increase in MRI signal that encompassed the frontal horn ependyma. After 13 weeks of treatment with pembrolizumab, the right frontal lesion continued to decrease in size, and there was no enhancement next to the left-side resection bed. A corresponding decrease in MRI signal in the right frontal horn was observed when compared with the previous signal. Immunohistochemistry (IHC) analysis of the resected primary tumors and metastases obtained before and three weeks after pembrolizumab treatment identified brisk CD3^+^, CD4^+^, and CD8^+^ T-cell infiltration in the post-treatment lesion.

In 2017, the first immunotherapy with pembrolizumab in metastatic CRC was reported [50]. An 81-year-old Hispanic male CRC patient was found to have PolE-V411L and RAF1-R256S mutations with a high TMB of 122 mutations per Mb. The tumor near the hepatic flexure significantly decreased in size after three and six cycles of treatment with 200 mg pembrolizumab every three weeks. IHC results showed large amounts of CD8^+^ tumor-infiltrating lymphocytes, >90% of which were PD-1 positive. Interestingly, >99% of PD-L1 expression occurred in the tumor microenvironment (TME).

In subsequent investigations, five patients with MSI-H and three patients with PolE-mutated (one *PolE-V411L* and two PolE-P286R) metastatic CRC were treated with a PD-1 inhibitor [51]. Four patients (three MSI-H and one *PolE*-mutated) responded to PD-1 blockade while the other four (two MSI-H and two *PolE*-mutated) were nonresponders. There were more CD8^+^ T cells, with the majority expressing PD-1, in the four responders than in nonresponders. In three out of the four responders, PD-1-expressing CD8^+^ T cells were mostly found in the tumor stroma. In addition, non-tumor PD-L1 expression was shown in CD68^+^ tumor-associated macrophages in all eight patients, whereas in only one responder with MSI-H, less than 5% of PD-L1 was observed on tumor cells. PD-L1 expression varied among individual responders (from ~600 to 3800 PD-L1 positive cells/cm^2^) and nonresponders (from ~400 to 3800 positive cells/cm^2^), but average levels of PD-L1 expression were not significantly different between the responder and nonresponder groups. Additionally, a large amount of CD4^+^ T-bet^+^ T-cells were detected within the TME of the responders as compared to that of the nonresponders.

Taken together, these clinical studies indicate that *PolE* mutations are associated with high TMB, abundant neoantigens, increased TILs in TME, and a good response to pembrolizumab treatment, as shown in Figure 1. Although there are increasing numbers of reported cases of cancer patients with *PolE* mutations showing good response to immune checkpoint blockade of PD-1 [38,49,50,51,53,54,55,56,57,58], some patients with *PolE* mutations still do not respond to the treatment. Investigating the mechanisms by which *PolE*-mutant tumors respond to immunotherapy would help identify ways to improve the efficacy.

## 7. Discussions 

Since *PolE*-associated ultramutated tumors were discovered in 2012, there has been significant progress in pathogenicity and clinical application. Work in yeast indicates that the arginine substitution at the 286th amino acid residue of the catalytic subunit of Polε prevents single strand DNA from entering the proofreading core and generating the hyperactive polymerase [22,24,25]. Subsequent studies in a *PolE-P286R* mouse model confirm the pathogenicity of *PolE* mutation in cancer [23]. Other *PolE*-associated mutations have also been explored for their role in ultramutagenesis [19,37]. However, several questions still remain. *PolE-V411L* mutants are distinct from those of *PolE-P286R*, as shown by structural and genetic data [19]. V411 is located outside of the binding cleft in the exonuclease domain. Although *PolE-V411L* is found in ultramutated tumors, surprisingly, *Pol2-V411L*, a yeast analog of *PolE-V411L*, does not generate elevated mutagenesis in yeast. It is still not understood how *PolE-V411L* is involved in ultramutagenesis. A possible mechanism could be that PolE V411L may alter the interaction between PolE and other essential proteins involved in DNA replication and/or DNA damage repair, leading to an increased mutation rate. Thirdly, although PolE-P286R and PolE-V411L may affect the behavior of DNA Polε differently, the mutation signatures of tumors harboring *PolE-P286R* and *PolE-V411L* are very similar [35]. The underlying reasons are not known.

*PolE*-driven ultramutated CRC and EC have good prognosis and favorable response to immunotherapy, possibly due to enhanced anti-tumor immune response caused by significantly elevated antigenic tumor neopeptides [47,59]. This is consistent with the observations in MSI-H CRC, in which increased numbers of neoantigens are correlated with enhanced immune infiltration and better patient survival [46]. Predicted neoantigen load in PolE-associated tumors is higher than in MSS and MSI-H tumors [60]. Notably, in MSI-H CRC, frameshift indels generate more neoantigens than single-nucleotide variants (SNVs) [46,61]. It is not clearly demonstrated which neoantigens contribute to increased immune infiltration and what percentage of neoantigens are generated by indels or SNVs in *PolE* and MSI tumors.

In addition to neoantigen load, another possible mechanism of the favorable outcome of PolE-mutant cancer patients is the “error threshold” theory. If too many mutations occur in the genome beyond an “error threshold”, cell viability would dramatically decrease. This phenomenon was proven in a yeast model with a combination of defective DNA polymerase proofreading and a complete loss of MMR [62,63]. Combining *Pol2-P301R* mutation and the deletion of *MSH6*, yeast cells did not develop after microcolonies were formed [24]. *MMR* deficient or *PolE*-mutant tumors rapidly accumulate large amounts of mutations (~600 mutations/cell division), reaching but not exceeding, ~20,000 exon mutations in under six months, suggesting a threshold compatible with cancer cell survival [64]. Moreover, the mutation rate in the *PolE P286R* mutant mouse cell line is estimated to be ~110 exon mutations per cell division, still not beyond the threshold of cancer cell survival [23]. More studies are needed to validate the “error threshold” mechanism.

## 8. Conclusions

In summary, cancer patients bearing *PolE* mutations display hypermutation and demonstrate good prognosis and a favorable outcome. A large amount of progress has been made in determining how *PolE* mutations affect DNA replication and mutation accumulation. However, most of the studies in human cancer are correlative. Understanding the mechanisms of *PolE* mutations in cancer biology would provide insight to a causal relationship between genome instability and cancer progression. It would also help to develop new strategies for effective cancer treatment.

## Figures and Tables

**Figure 1 cancers-14-01467-f001:**
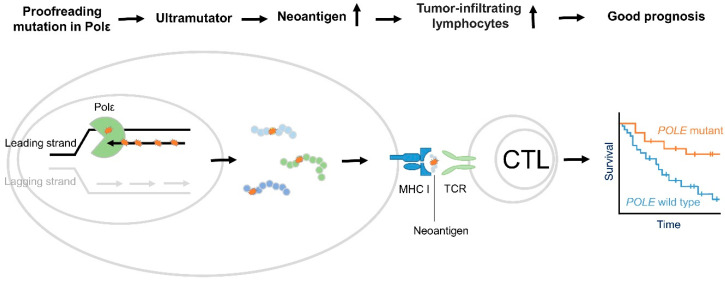
Proposed mechanism of good prognosis in *PolE* associated cancer. *PolE* encodes the catalytic and proofreading exonuclease subunit of DNA polymerase ε (Pol ε), which is responsible for bulks of DNA synthesis in the leading strand during DNA replication. Cancer-associated PolE mutations in the exonuclease domain hinder proofreading activity and increase polymerase activity, resulting in ultramutation [22,23,24,25]. Subsequently, they lead to an increased neoantigen load [44,47]. When neoantigens are presented, the immune system recognizes them and ultimately generates an immune response, eliciting tumor suppressor function [44,47].

**Table 1 cancers-14-01467-t001:** Immunotherapy in PolE mutant cancers.

PolE Mutation	Cancer Type	Tumor Mutation Burden	Immunotherapy	Overall Best Response	Reference
somatic V411L	metastatic EC	4500 to 6500 nonsynonymous mutations	pembrolizumab	PR	[49]
germline V424L	glioblastoma	17,276 to 20,045 mutations	pembrolizumab	PR	[38]
somatic V411L	metastatic CRC	122 mutations per Mb	pembrolizumab	PR	[50]
somatic V411L	metastatic CRC	NA	pembrolizumab	CR	[51]
somatic P286R	metastatic CRC	NA	pembrolizumab	PD	[51]
somatic P286R	metastatic CRC	NA	pembrolizumab	SD	[51]

EC, Endometrial Cancer; CRC, Colorectal Cancer; PR, Partial Response; CR, Complete Response; PD, Progressive Disease; SD, Stable Disease.

## Data Availability

No new data were created or analyzed in this study. Data sharing is not applicable to this article.

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
