# Peer review of "Polymerase Epsilon-Associated Ultramutagenesis in Cancer"

_cancers, 2022, doi:10.3390/cancers14061467_

Round 1
Reviewer 1 Report
The manuscript "Polymerase Epsilon-Associated Ultramutated Cancer: A Systematic Review" by Xing et al is a very comprehensive review article in the field. The authors have covered most of the literature in the field and the review covers some interesting aspects of POLE mutated cancers and immunotherapeutic options.
The only comments I have regarding improving the review are:
- The authors cite Alexandrov et al (2013) manuscript describing mutation signatures. Since then, a newer study with better signature assignment in tumors was released in 2020. The authors should update the citation accordingly.
- Page 3-4 - The authors discuss the role of POLE variants in driving carcinogenesis by mutating cancer driver genes. Various studies from Jason Wong's laboratory have demonstrated a role of POLE in driving mutations in cancer drivers. The authors may want to discuss these findings as well in the review. PMIDs - 30412573, 32012149.
- The authors are also missing a citation of a study by Ian Tomlinson's lab on the timing of POLE mutations in tumors and their role in the production of clonal neoantigens. The authors may want to also discuss this study - PMID 29604063.
Author Response
Reviewer 1:
1. The authors cite Alexandrov et al (2013) manuscript describing mutation signatures. Since then, a newer study with better signature assignment in tumors was released in 2020. The authors should update the citation accordingly.
We have revised the sentences on p3 and cited the new study (ref. 36) in the manuscript (lines 143-147).
2. Page 3-4 - The authors discuss the role of POLE variants in driving carcinogenesis by mutating cancer driver genes. Various studies from Jason Wong's laboratory have demonstrated a role of POLE in driving mutations in cancer drivers. The authors may want to discuss these findings as well in the review. PMIDs - 30412573, 32012149.
Accordingly, we have added the sentence “Another study found that truncated TP53 R213* was specifically enriched in PolE P286R mutant CRC [37]” on p4, lines 184-185.
3. The authors are also missing a citation of a study by Ian Tomlinson's lab on the timing of POLE mutations in tumors and their role in the production of clonal neoantigens. The authors may want to also discuss this study - PMID 29604063.
Accordingly, “It has been shown that the PolE mutations are identified in pre-malignant lesions in EC and CRC with prominent CD8+ T-cell infiltration. It suggests that somatic PolE mutations are early events and may contribute to tumor initiation and good prognosis [46]” has been added on p5, lines 207-209.
Reviewer 2 Report
In this manuscript, the authors presented a review of PolE-associated tumors with ultramutation genotype and its correlation with clinical prognosis. They focus mainly on P286R and V411L mutations in PolE found in cancer cells which are also the best characterized among PolE mutations found in cancer cells. Mutations in the leading-strand DNA polymerase PolE are located mainly in the protein’s exonuclease region, resulting in impaired functioning of the proofreading mechanism of this polymerase. This results in ultramutated tumors generating increased neoantigen load leading to an immune response. An alternative “error-threshold” theory is also discussed in this review.
The manuscript is clearly written and brings together interesting clinical and experimental data. I only have some minor comments presented below:
- Among the two most common somatic mutations in PolE P286R is located in the exonuclease motif, which explains high mutation rates observed in this mutant, while V411L is located beyond the exonuclease motif. As the authors say, “it is still not understood how PolE-V411L is involved in ultramutagenesis”. However, maybe the authors could speculate on the nature of PolE-V411L role in increasing the mutation rates.
- The authors discuss mainly the effects of mutations in the gene encoding the catalytic subunit of Polymerase Epsilon. In line 325 they mention a mutation in PolE4, a non-catalytic subunit of this polymerase. A comment on the involvement of mutations in other non-catalytic subunits of Pol Epsilon could be valuable.
- I could not find any reference to Figure 1 in the text.
Author Response
Reviewer 2:
1. Among the two most common somatic mutations in PolE P286R is located in the exonuclease motif, which explains high mutation rates observed in this mutant, while V411L is located beyond the exonuclease motif. As the authors say, “it is still not understood how PolE-V411L is involved in ultramutagenesis”. However, maybe the authors could speculate on the nature of PolE-V411L role in increasing the mutation rates.
Accordingly, “Possible mechanisms could be that PolE V411L may alter the interaction between PolE and other essential proteins involved in DNA replication and/or DNA damage repairs, leading to increased mutations” has been added on p9.
2. The authors discuss mainly the effects of mutations in the gene encoding the catalytic subunit of Polymerase Epsilon. In line 325 they mention a mutation in PolE4, a non-catalytic subunit of this polymerase. A comment on the involvement of mutations in other non-catalytic subunits of Pol Epsilon could be valuable.
Accordingly, “Mutations in non-catalytic subunits of PolÉ›, PolE2, PolE3 and PolE4, are rare in cancer and their involvement in ultramutagenesis is not clear” was added on p3, lines 99-100.
3. I could not find any reference to Figure 1 in the text.
We have added references in the figure legend.